# Blockade of CDK7 Reverses Endocrine Therapy Resistance in Breast Cancer

**DOI:** 10.3390/ijms21082974

**Published:** 2020-04-23

**Authors:** Yasmin M. Attia, Samia A. Shouman, Salama A. Salama, Cristina Ivan, Abdelrahman M. Elsayed, Paola Amero, Cristian Rodriguez-Aguayo, Gabriel Lopez-Berestein

**Affiliations:** 1Pharmacology Unit, Cancer Biology Department, National Cancer Institute, Cairo University, Kasr Al Eini Street, Fom El Khalig, Cairo 11796, Egypt; yasmin.mostafa@nci.cu.edu.eg (Y.M.A.); samia.shouman@nci.cu.edu.eg (S.A.S.); 2Department of Experimental Therapeutics, The University of Texas MD Anderson Cancer Center, Houston, TX 77030, USA; CIvan@mdanderson.org (C.I.); AMHamoda@mdanderson.org (A.M.E.); PAmero@mdanderson.org (P.A.); CRodriguez2@mdanderson.org (C.R.-A.); glopez@mdanderson.org (G.L.-B.); 3Pharmacology & Toxicology Department, Al-Azhar University, Cairo 11675, Egypt; 4Center for RNA Interference and Non-Coding RNA, The University of Texas MD Anderson Cancer Center, Houston, TX 77030, USA; 5Department of Cancer Biology, The University of Texas MD Anderson Cancer Center, Houston, TX 77030, USA

**Keywords:** breast cancer, estrogen receptor, resistance, tamoxifen, cyclin-dependent kinase 7, c-Myc

## Abstract

Cyclin-dependent kinase (CDK)-7 inhibitors are emerging as promising drugs for the treatment of different types of cancer that show chemotherapy resistance. Evaluation of the effects of CDK7 inhibitor, THZ1, alone and combined with tamoxifen is of paramount importance. Thus, in the current work, we assessed the effects of THZ1 and/or tamoxifen in two estrogen receptor-positive (ER+) breast cancer cell lines (MCF7) and its tamoxifen resistant counterpart (LCC2) in vitro and in xenograft mouse models of breast cancer. Furthermore, we evaluated the expression of CDK7 in clinical samples from breast cancer patients. Cell viability, apoptosis, and genes involved in cell cycle regulation and tamoxifen resistance were determined. Tumor volume and weight, proliferation marker (Ki67), angiogenic marker (CD31), and apoptotic markers were assayed. Bioinformatic data indicated CDK7 expression was associated with negative prognosis, enhanced pro-oncogenic pathways, and decreased response to tamoxifen. Treatment with THZ1 enhanced tamoxifen-induced cytotoxicity, while it inhibited genes involved in tumor progression in MCF-7 and LCC2 cells. In vivo, THZ1 boosted the effect of tamoxifen on tumor weight and tumor volume, reduced Ki67 and CD31 expression, and increased apoptotic cell death. Our findings identify CDK7 as a possible therapeutic target for breast cancer whether it is sensitive or resistant to tamoxifen therapy.

## 1. Introduction

Breast cancer is one of the most common cancers and the second leading cause of cancer death in female patients [1]. Approximately 70% of all breast cancers are hormone-dependent and express estrogen receptor alpha (ER-α) [2]. Endocrine therapy is effective in approximately 60% of hormone receptor-positive breast cancer cases [3]. Tamoxifen, a selective ER modulator, is the most commonly used neoadjuvant and adjuvant endocrine therapy in treatment of patients with ER-α+ breast cancer [4], tamoxifen targets the proliferation-stimulating effects of ERs in breast cancer [5]. Despite obvious benefits of tamoxifen in the treatment and/or chemoprevention of breast cancer, most of the initially responsive breast tumors experience a recurrence due to the development of acquired tamoxifen resistance [6].

Multiple mechanisms are implicated in the development of tamoxifen resistance, including mutations of the ERs; activation of tyrosine kinases such as HER2 and insulin growth factor receptor [6]; and alterations in the cell cycle. A deeper understanding of the underlying mechanisms of resistance to endocrine therapy will facilitate the development of new strategies for the treatment of patients with breast cancer.

The cyclin-dependent kinases (CDKs) are a family of protein kinases that are involved in the regulation of the cell cycle and gene transcription. Cyclin-dependent kinases play fundamental roles in cancer development and metastasis [7]. Cyclin-dependent kinases remain inactive until they bind to cyclins which are regulatory proteins [8]. Then, the CDK-cyclin complex phosphorylates its substrates on serine and threonine sites [9]. The human genome contains 21 genes encoding CDKs and five additional genes encoding a more distant group of proteins known as CDK like (CDKL) kinases [10]. Cyclin dependent kinase-7 CDK7 has a critical role in gene transcription by functioning as a component of the general transcription factor II Human (TFIIH) [11]. In addition, CDK7 can promote cell cycle progression by acting as a CDK-activating kinase (CAK) in a trimeric complex with Cyclin H and MAT1, phosphorylating cell cycle CDKs such as CDK1, CDK2, CDK4, and CDK6 [12]. Thus, targeting CDK7 offers a unique strategy to target transcription and cell cycle progression simultaneously. CDK7 is highly expressed in breast cancer and is associated with poor prognosis and is correlated with poor response to endocrine treatment [13]. CDK7 stimulates the phosphorylation of serine 118 of the ER, which is a crucial step governing ER activity, degradation, and regulation [14]. CDK7 is also a master regulator of important genes in cancer, as it regulates MYC expression in many cancer types [15,16]. In breast cancer patients, high MYC expression is associated with endocrine therapy resistance and poor survival outcomes [17]. Thus, we hypothesized that CDK7 contributes to tamoxifen resistance and that CDK7 inhibition can enhance the cytotoxic effect of tamoxifen, particularly in tamoxifen-resistant breast cancer.

## 2. Results

### 2.1. Increased Cyclin-Dependent Kinase (CDK7) Expression Correlates with Poor Patient Survival and Tamoxifen Resistance in Estrogen Receptor + (ER+) Breast Cancer

We first examined mRNA sequencing data from The Cancer Genome Atlas (TCGA) database for patients with breast cancer to assess the potential correlation between *CDK7 and estrogen receptor alpha or 1 (ER1 or ER-α)* levels. We found a positive correlation between *CDK7* and *ER1* mRNA levels in TCGA breast cancer samples (Figure 1a) (rho = 0.41; *p* = 0.02) and ER+ breast cancer samples (rho = 0.27; *p* < 0.001) (Figure 1b). High expression of *CDK7* (first sextile) was associated with significantly shorter overall survival (OS) for only ER+ breast cancer patients (*p* = 0.01). Cox proportional hazards regression analysis of OS yielded a univariate hazard ratio (HR) of 2.64 (95% CI, 2.33–5.22; *p* = 0.005) and a multivariate HR (adjusted for age and *CDK7* expression) of 2.7 (95% CI, 1.24–5.92; *p* = 0.012) (Figure 1c). We next analyzed microarray data for a cohort of breast cancer patients receiving tamoxifen. The median OS duration was 68.5 months for patients with high *CDK7* expression and not reached for patients with low *CDK7* (Figure 1d). The Cox regression analysis of OS yielded an HR of 1.5 (95% CI, 1.15–1.96; *p* = 0.0028). Because CDK7 is a master regulator oncogenes expression, such as MYC, we next explored the relationship between CDK7 and MYC expression. Our data revealed a significant correlation between *CDK7* and *MYC* expression. We found a significant correlation (rho = 0.41; *p* = 0.02) between CDK7 and *MYC* expression in patients with ESR+ breast cancer treated with tamoxifen (Figure 1e). Furthermore, we found that median OS durations in patients with ER+ breast cancer receiving tamoxifen were longer when *MYC* expression levels (where the cutoff point is 0.71) were lower (median OS, 79.8 months for high *MYC* expression group and not reached for low *MYC* expression group; HR, 1.49; 95% CI, 1.07–2.08; *p* = 0.0177) (Figure 1f). Finally, as shown in Figure 1g, we found that MYC expression correlates positively with ER1 expression in breast cancer patients receiving tamoxifen (rho = 0.46, *p* < 0.0001)

### 2.2. Targeting CDK7 Decreases Estrogen Receptor (ER) Activation

The results of the TCGA analysis prompted us to examine the relationship between CDK7 and ER-α expression in tamoxifen-sensitive and -resistant cell lines. To establish the in vitro working model, we screened the CDK7 expression level in the tamoxifen-sensitive MCF-7 cell line and its tamoxifen resistant counterpart LCC2 cells. We found that CDK7 expression levels in LCC2 cells were higher than those in MCF-7 cells (Figure 2a). 

It has been reported that CDK7 plays a role in the phosphorylation of ER-α, specifically at Ser118 [18]. Thus, we next determined the levels of pan-ER-α and phosphorylated ER-α in tamoxifen-sensitive and -resistant cells by Western blotting. The ratio of phosphorylated ER-α (Ser118) to total ER-α was 1.5 times higher in tamoxifen-resistant LCC2 cells than in tamoxifen-sensitive MCF-7 cells (Figure 2b). Next, we determined the effect of CDK7 inhibition in MCF-7 and LCC2 cells. To inhibit CDK7, we used two different approaches: the small molecule inhibitor THZ1, which covalently binds to CDK7 and suppresses its kinase activity and its expression with high selectivity and smalls interfering RNAs (siRNAs) that specifically target and degrade CDK7 [18,19]. We analyzed the effect of treatment with THZ1 or two different siRNAs on CDK7 levels by Western blotting and real-time polymerase chain reaction (RT-PCR). Our data revealed that transfection with CDK7 siRNA (at concentrations of 50 and 100 nM) reduced CDK7 protein levels (Figure 2c,d) and CDK7 mRNA levels (Figure 2e,f) in MCF-7 and LCC2 cells. However, treating cells with tamoxifen did not affect the level of CDK7 in either cell line. Next, levels of total ESR-α and phosphorylated ER-α in cells transfected with CDK7-targeting siRNAs or treated with THZ1 were analyzed by Western blotting. In tamoxifen-sensitive MCF-7 and tamoxifen-resistant LCC2 cells, siRNA-CDK7-1 at a concentration of 100 nM, but not 50 nM, decreased the ratio of phosphorylated ESR-α (Ser118) to total ESR-α to half of the level in control siRNA-treated and untreated cells. However, siRNA-CDK7-2 at concentrations of both 50 and 100 nM reduced the ratio of phosphorylated ESR-α (Ser118) to total ER-α compared with control cells. Similarly, treatment of either MCF-7 cells or LCC2 cells with THZ1 resulted in a marked reduction of phosphorylated ESR-α/total ESR-α ratio (Figure 2g, h). Since the transfection with CDK7- siRNA-2 resulted in a more pronounced knockdown of CDK7, we used it rather than siRNA-CDK7-1 in subsequent experiments.

Given the pivotal role of CDK7 in regulating the expression of genes involved in oncogenesis, stemness, tumor growth, and progression, we next determined the effect of CDK7 inhibition on the expression of MYC, STAT3, and β-catenin. Our resulted revealed that inhibition of CDK7 with THZ1 or siRNA decreased MYC, STAT3, β-catenin, and TFIIH expression in both MCF-7 and LCC2 cells (Figure 2i,j). Taken together, our results demonstrated that inhibition of CDK7 abolished the activation of ER-α and the expression of MYC and STAT3. 

### 2.3. Targeting CDK7 Augmented Tamoxifen-Induced Cytotoxicity

Given the possible role of CDK7 in oncogenesis, stemness, and tumor growth, we next examined whether inhibition of CDK7 by siRNA or THZ1 affected tumor cell growth by using a sulforhodamine B (SRB) assay. As indicated in Figure 3a, treatment of MCF-7 and LCC2 cells with tamoxifen for 48 h resulted in a dose-dependent growth inhibition in both cell lines. The half maximal inhibitory concentrations (IC_50_) of tamoxifen in MCF-7 and LCC2 cell lines were 6.5 µM and 67.7 µM, respectively. Treatment with THZ1 demonstrated similar effects in MCF-7 and LCC2 cell lines. Thus, the IC_50_ values of THZ1 in MCF-7 cells and LCC2 cells were 11 and 13 nM, respectively (Figure 3b). Two different siRNAs against CDK7 both decreased the viability of MCF-7 and LCC2 cells with similar magnitude (Appendix A). 

Next, the concentrations of tamoxifen that caused 10% cell growth inhibition (IC_10_) or 25% cell growth inhibition (IC_25_) in MCF-7 cells were determined (1 µM and 3.5 um, respectively) and were combined with IC_10_ of THZ1 (1 nM). As shown in Figure 3c, the combination of tamoxifen and THZ1 inhibited cell growth more effectively than did either treatment alone. More importantly, in tamoxifen-resistant LCC2 cells, the combination of IC_10_ (17 µM) or IC_25_ (35 µM) of tamoxifen with IC_10_ (1 nM) of THZ1 decreased cell viability by more than 50%. In both cell lines, silencing CDK7 with siRNA also inhibited cell growth more effectively when combined with tamoxifen than did either treatment alone (Figure 3d). To determine whether the duration of treatment affected the combination effect, cells were treated for 24, 48, or 72 h and cell viability was assessed. As demonstrated in Appendix A, there were no differences in either cell line. 

The exposure of cells to cytotoxic agents can reduce cell viability or increase cell death through necrosis or apoptosis. A common characteristic of both types of cell death is damage to the cell membrane which results in the release of the cytoplasmic enzyme lactate dehydrogenase (LDH) to the extracellular space. Therefore, we determined the level of LDH in the media to confirm the cytotoxicity of these combinations. The combination of 1 nM THZ1 with 1 µM tamoxifen in MCF-7 or 17 µM tamoxifen in LCC2 increased the levels of LDH in the media more than did tamoxifen or THZ1 alone (Appendix A). 

### 2.4. CDK7 Inhibition Increased Tamoxifen-Induced Apoptosis and Modulated Its Effect on the Cell Cycle

Since CDK7 has a known role in cell cycle regulation, we analyzed the effect of treatment with CDK7 inhibitor THZ1 and/or tamoxifen on the cell cycle and apoptosis. The cell cycle analysis in MCF-7 cells (Appendix A) showed that treatment with tamoxifen (1 µM) caused G1 arrest, while treatment with THZ1 (1 nM) or a combination of THZ1 and tamoxifen triggered G2 arrest. Additionally, CDK7 siRNA (50 nM), both separately and combined with tamoxifen, produced G1 arrest (Figure 4a). The different treatments showed the same effects in LCC2 cells (Appendix A): tamoxifen (17 µM) increased G1 arrest, while THZ1 (1 nM) alone or in combination with tamoxifen promoted G2 phase arrest. CDK7 siRNA, both alone and in combination with tamoxifen, also caused G1 arrest (Figure 4b). 

Annexin V assays were performed to show the effect of combining tamoxifen with CDK7 inhibition on apoptosis. Our resulted demonstrated that the combinations of tamoxifen with THZ1 or CDK7 siRNA decreased the number of viable MCF-7 and LCC2 cells more than did any of the treatments alone (Figure 4a,b, right panel). However, the percentage of apoptotic cells were the same in the cells treated with THZ1 and tamoxifen as compared to cells treated with either THZ1 or tamoxifen alone. Moreover, Western blotting for cell cycle markers showed that adding THZ1 or CDK7 siRNA to tamoxifen in both MCF-7 and LCC2 cells increased the levels of cleaved PARP and caspase-3 levels but no significant difference when compared to either treatment alone as shown (Figure 4c). This finding might indicate that THZ1 increased tamoxifen’s cytotoxic effect might be through mechanisms other than apoptosis. Cyclin D1 levels were decreased upon treatment with THZ1 (1 nM) or siRNA (50 nM) alone or in combination with tamoxifen (1 or 17 µM in MCF-7 and LCC2, respectively). In MCF-7 and LCC2 cells, THZ1 treatment alone and in combination with tamoxifen reduced cyclin E protein expression. However, the combination of CDK7 siRNA and tamoxifen decreased cyclin E levels only in MCF-7 cells. Cyclin H levels were reduced in MCF-7 cells treated with tamoxifen, CDK7 siRNA, and their combinations but not in those treated with THZ1. Conversely, in LCC2 cells, cyclin H levels decreased only with THZ1 treatment. Cyclin A1 levels increased after THZ1 in LCC2 cells only while CDK7 siRNA increased its level in both MCF-7 and LCC2 cells. On the other hand, tamoxifen alone or its combination with THZ1 or CDK7 siRNA decreased cyclin A1 level in both cell lines (Figure 4c). Cyclin B1 levels decreased in both cell lines exposed to most treatments, except for MCF-7 cells treated with CDK7 siRNA, in which levels did not differ from controls (Figure 4c). In MCF-7 cells, P^27^ and P^21^ levels were decreased in both cell lines after treatment with THZ1, THZ1 plus tamoxifen, and CDK7 siRNA plus tamoxifen. In LCC2 cells, the combination of CDK7 siRNA plus tamoxifen did not affect levels of P^27^ or P^21^ (Figure 4c) Although CDK4 protein levels were reduced by tamoxifen and by the combination treatments in MCF-7 cells, these levels were unchanged by tamoxifen treatment in LCC2 cells but were decreased by THZ1 its combination with tamoxifen. In MCF-7 cells, CDK2 levels were decreased in cells treated with tamoxifen, CDK7 siRNA, or tamoxifen plus THZ1 or CDK7 siRNA, whereas in LCC2 cells, CDK2 levels were decreased only in cells treated with THZ1. Finally, CDK6 was expressed at low levels in all treatment groups in MCF-7 cells except for tamoxifen-treated cells, while in LCC2 cells it was decreased in all groups (Figure 4c). 

In brief, although inhibition CDK7 and treatment with tamoxifen had variable effects on cell cycle markers, their combination decreased protein levels of many of these markers. These findings emphasize the potent effect of the THZ1 and tamoxifen combination on both tamoxifen-sensitive and tamoxifen-resistant breast cancer cell lines. Interestingly, inhibition of CDK7 by THZ1 had a different effect on the cell cycle from that of CDK7 inhibition by siRNA which suggests that THZ1 may act on the cell cycle via a mechanism other than inhibition of CDK7.

### 2.5. CDK7 Inhibition Enhances the Cytotoxic Effect of Tamoxifen in Murine Models of Tamoxifen-Sensitive and Tamoxifen-Resistant Breast Cancer

On the basis of our in vitro findings, we next examined the antitumor activity of the combination of THZ1 and tamoxifen on orthotopic murine models of tamoxifen-sensitive (MCF-7) and tamoxifen-resistant (LCC2) breast cancer. To generate the orthotopic models, estrogen pellets were implanted in the mice 1 week before inoculation of MCF-7 or LCC2 cells into the mammary fat pad. After 7 days, mice were randomized into four treatment groups (10 mice per group) (Appendix A) and received no treatment, tamoxifen, THZ1, or the combination of tamoxifen plus THZ1.

In the xenograft model using MCF-7, after 6 weeks of treatment with tamoxifen, THZ1 or tamoxifen combined with THZ1, tumor volume was reduced to 25%, 45%, and 14%, respectively, as compared to the untreated control group (*p* < 0.05) (Figure 5a). Similarly, the tumor weight was reduced to 50%, 47%, and 10% in tamoxifen, THZ1, and combination-treated mice compared to untreated control group (Figure 5b). Mouse weight did not differ significantly among treatment groups which suggests that there were no toxic effects from the treatments (Appendix A). 

c-MYC (Cellular MYC Proto-Oncogene, BHLH Transcription Factor) is one of the most upregulated oncogenes in several types of cancer and has been reported to play variable roles in different molecular subtypes of breast cancer [20]. A previous study has shown increased c-MYC expression in 28 pairs of primary and metastatic tumors from patients treated with tamoxifen [21]. This finding could explain the role of estrogen in regulating cell growth via c-Myc which in turn regulates 50% of all acutely estrogen-regulated genes and makes up 85% of the cell growth signature [22]. To explore the relationship between CDK7 and MYC expression in vivo, Western blotting of tumor tissue lysates was performed. In agreement with our in vitro results, THZ1 alone or combined with tamoxifen decreased MYC expression in MCF-7 tumors, while tamoxifen alone increased MYC levels (Figure 5c). Given our previous results, we performed immunohistochemical analyses for CD31 and Ki67 and terminal deoxynucleotidyl transferase dUTP nick end labeling (TUNEL) experiments to study the effect of combined treatment with THZ1 and tamoxifen on angiogenesis, cell proliferation, and apoptosis, respectively. Tumors from mice treated with tamoxifen and THZ1 showed significantly reduced cell proliferation (*p* = 0.0001) (Figure 5d). Because CDK7 is necessary for the expression of MYC, which is important for vasculogenesis and angiogenesis throughout tumor development and progression, we also studied the effects of CDK7 inhibition on angiogenesis. The THZ1 treated group had a significantly reduced microvessel density compared with controls (*p* = 0.001). Analysis of TUNEL data (Figure 5d) showed a significant increase in apoptosis induction in tumor treated with tamoxifen alone (6 fold increase, *p* < 0.01), THZ1 alone (12 fold increase, *p* < 0.01); and even higher rates of apoptosis were observed in the tamoxifen plus THZ1 combination group (20-fold increase, *p* < 0.001).

With regard to the murine model of established from LCC2, similar pattens of the effect of THZ1 and/or tamoxifen were observed. Thus, treatment with tamoxifen or/and THZ1 reduced the tumor volume to 68% and 54%, 38%, respectively (*p* < 0.05) (Figure 5e). Moreover, tumor weight decreased to 80%, 71%, and 50% in tamoxifen, THZ1, and combination treatment group, respectively (*p* < 0.05) compared to the untreated group (Figure 5f). Mouse weight did not differ between groups, indicating that there were no toxic effects from the treatments (Appendix A). 

As in the tamoxifen-sensitive model, inhibition of CDK7 in LCC2 tumors decreased MYC levels on Western blotting (Figure 5g). Immunohistochemistry and TUNEL experiments showed that LCC2 tumors from mice treated with tamoxifen plus THZ1 had an 80% reduction in cell proliferation (*p* < 0.0001) (Figure 5h). Tumors from mice in the THZ1 treatment group had significantly lower microvessel density than did controls (50% reduction, *p* < 0.0001), while tumors in the combination treatment group had 80% lower microvessel density (*p* < 0.0001). TUNEL data (Figure 5h) showed a significant increase in apoptosis induction in mice treated with THZ1 alone (13 fold increase, *p* < 0.01) or with THZ1 plus tamoxifen (20 fold increase, *p* < 0.001). 

Taken together, our in vivo results supported our in vitro and TCGA findings, that by targeting CDK7, it enhances the cytotoxic effect of tamoxifen in both tamoxifen-sensitive and tamoxifen-resistant cells. These findings allowed us to illustrate the mechanism by which targeting CDK7 enhances tamoxifen toxicity and overcomes tamoxifen resistance (Figure 6).

## 3. Discussion

Our findings show that cyclin dependent kinase (CDK7) is a major factor in tamoxifen resistance. Indeed, CDK7 was found to be associated with the important transcription factor MYC and with the STAT3 and β-catenin pathways which play crucial roles in tamoxifen resistance, stemness, and cancer progression. 

Estrogen receptor (ER) is a key regulator that is expressed in almost 70% of breast cancer cases [2,23]. In a very recent study, a positive correlation was found between CDK7 and survival in both luminal B (ER+) and HER+ [12]. CDK7 is one of the first CDKs to be involved in Pol II transcription cycle. CDK7 phosphorylates the RNA polymerase II (pol II) C-terminal domain (CTD) and activates the P-TEFb-associated kinase CDK9. In addition, inhibition of CDK7 reduced capping enzyme recruitment, increased pol II promoter-proximal pausing, and defective termination at gene 3’ ends. [11]. In the current study, we showed a correlation between CDK7 levels and ER-α expression in all breast cancers and in ER+ breast cancers in particular. Additionally, high CDK7 expression was correlated with shorter OS in patients with ER+ breast cancer, especially those treated with tamoxifen. These results indicate that CDK7 may be involved in resistance to endocrine therapy. Indeed, our in vitro studies showed higher levels of CDK7 in tamoxifen-resistant cells than in tamoxifen-sensitive cells. Moreover, targeting CDK7 with a small molecule inhibitor or siRNAs decreased ER phosphorylation at Ser118 in tamoxifen-sensitive and tamoxifen-resistant cells. Ser118 phosphorylation has been linked with poor response to endocrine therapy [24] and implicated in resistance to endocrine therapy [25,26]. A retrospective clinical study that included more than 100 female patients showed that phosphorylation of ERα at Ser118 was correlated with tamoxifen resistance, HER2 overexpression, and poor prognosis [27]. This association of CDK7 with ERα activation suggested that the combination of CDK7 inhibition with endocrine therapy may be a promising strategy for overcoming resistance to endocrine therapy. 

Dysregulation of the cell cycle, upregulation of cyclins and CDKs, and downregulation of cell cycle checkpoints are common characteristics of cancer cells [28,29]. Therefore, targeting cell cycle components should be more effective on cancer cells than normal cells. Currently many CDK inhibitors are in clinical use, including the CDK4/6 inhibitor palbociclib, which showed a synergistic effect with endocrine therapy in patients with ERα+ breast cancer [30,31]. CDK7 regulates the cell cycle by activating CDK1, 2, 4, and 6 [19]. Additionally, our results showed that THZ1 produced cell cycle arrest in G2/M phase which is in agreement with previous study that reported that, THZ1 significantly inhibited proliferation of non-small-cell lung cancer (NSCLC) cells and arrested the cell cycle in G2/M phase along with downregulation of cyclins and CDKs [32]. Moreover, knockdown of CDK7 with siRNA resulted in a different pattern on cell cycle (G1 arrest). This differential effect might contribute to pleiotropic effect of THZ1. Thus, it is speculated that CDK7 inhibitor such as THZ1 inhibit cell proliferation and survival through multiple mechanisms. In this regard, the CAK-inhibitory activity of THZ1 [33] was associated with diminished phosphorylation/activation of CDK9, as well as CDK1 and 2 and arrest cell cycle progression in S-phase as well as G2/M arrest [32]. Additionally, tamoxifen-induced G1 arrest in both MCF-7 and LCC2 cells is similar to the findings of a previous study in which the percentage of cells at G0/G1 phase was markedly increased and the percentage of cells at S and G2/M phases were decreased, respectively [34]. The combination of THZ1 with tamoxifen caused G2 arrest and decreased expression of G1 and M phase cyclins (including D1, E1 and B1) and the G1 acting CDKs;, CDK2, 4 and 6. Interestingly, the expression of CDK inhibitors (CDKIs) was lower in the group treated with THZ1 and tamoxifen compared with untreated cells. When tamoxifen was combined with CDK7 siRNA, it caused G0/G1 phase arrest. However, the expression of G1 cyclins (including D1, E1) was decreased in breast cancer cells. Moreover, G2 and M phase cyclins (A1, B1) and G1 acting CDKs; CDK2, 4, and 6 were decreased in the combination-treated group. The expression of CDKIs levels (p21 and p27) were reduced in MCF-7 and remained the same in LCC2 cells. CDK7 inhibitors combined with tamoxifen reduced the expression level of cyclin D1 and E1 which were reported to be associated with tamoxifen resistance [35,36]. Also, CDK7’s role in overcoming chemotherapeutic resistance in breast cancer was mentioned before, where CDK7 inhibition by TZH1 was found to reverse the resistant in HER2+ breast cancer patients [37]. Moreover, CDK7 inhibition increased tamoxifen-induced apoptosis in vitro and in vivo models compared to inhibitor or tamoxifen individually. However, the combination did not show any difference from THZ1 or siRNA alone on the level of caspses-3 and PARP. In a previous study, THZ1 showed an increase in apoptosis in different cancer types [38,39].

CDK7 is overexpressed in endocrine therapy-resistant tumors [18]. Here, we report that in a cohort of patients receiving tamoxifen as a hormonal therapy for breast cancer, patients with a high level of CDK7 had a lower survival rate than did those with low CDK7. Moreover, CDK7 inhibition has been shown to be effective in low concentrations in many cancer types, including leukemia [19], ovarian cancer [40], and colon cancer [41], which makes CDK7 inhibitors good candidates for use as adjuvant breast cancer treatment. 

CDK7 is a master regulator of gene transcription that plays essential roles in transcription initiation and elongation by phosphorylating RNA polymerase II. Thus, knockdown of CDK7 silences many genes [42,43]. One of the genes that CDK7 regulates is MYC (MYC Proto-Oncogene, BHLH Transcription Factor) [44]. MYC is an estrogen-regulated oncogene [45] that is overexpressed in many cancer types including breast cancer [41,42]. Previous studies have shown that ER activates MYC expression and that MYC may modulate estrogen-mediated signaling [46,47]. In the present study, we found that MYC expression was correlated with ER-α expression in patients receiving tamoxifen and that high MYC expression was associated with shorter OS in those patients. MYC has also been reported to regulate the expression of angiogenic factors, such as angiopoietin-2, and of angiogenic inhibitors such as thrombospondin-1 [48]. Studies have also shown a correlation between MYC expression and that of VEGF, a key player in physiological and pathological angiogenesis [49]. Our findings regarding MYC’s effect on patient survival and on CDK7 expression were expected, as upregulation of MYC enhances the expression of key elements of the cell cycle such as cyclin D, cyclin E, and the E2F family of transcription factors; MYC also regulates CDK7 phosphorylation [50,51]. MYC is overexpressed in patients with ER+ breast cancer that is resistant to tamoxifen; this may be related to its roles in the cell cycle, oncogenesis, and tumor progression [52]. In the present work, we demonstrated that targeting CDK7 with a small molecule inhibitor, THZ1, or with siRNAs decreased the level of MYC in tamoxifen-sensitive and tamoxifen-resistant in vitro and *in vivo models*. Furthermore, inhibition of CDK7 decreased STAT3 and β-catenin levels in both tamoxifen-sensitive and -resistant cell lines [53]. Since upregulation of Wnt/β-catenin activity has been reported to enhance MYC expression [49,50], consequently decreasing both β-catenin and STAT3—possibly a way to overcome tamoxifen resistance.

Our recommendation for future studies includes using other endocrine therapy combinations with THZ1 or siRNA against CDK7 to confirm the role of CDK7 inhibition in overcoming endocrine resistance. More mechanistic studies should be conducted to explore the downstream targets of CDK7. Additional animal models should be used with siRNA separately or combined with tamoxifen in sensitive and resistant cells, as well as different CDK7 inhibitors in other types of breast cancer to explore if it has any significant effect. 

In summary, our mechanistic studies (summarized in Figure 6) provide evidence that CDK7 activates ER-α phosphorylation at Ser118 which enhances the transcription of MYC in ER+ breast cancer (Figure 6, pathway 1). Targeting CDK7 via siRNA or THZ1 could block ER activation and expression of ER target genes, including MYC (Figure 6, pathway 2) which may, in turn, overcome tamoxifen resistance. 

Endocrine resistance remains a challenge to the management of ER+ breast cancer. Although many mechanisms are involved in endocrine resistance, the cell cycle and cyclins are crucial factors in the development of endocrine resistance. In our study, we found that targeting CDK7 increased the toxic effect of tamoxifen in sensitive and resistant breast cancer cells as well as in vivo models by targeting important proteins, such as ER and MYC, and essential pathways in breast cancer progression (STAT3 and β-catenin).

## 4. Materials and Methods 

### 4.1. TCGA Breast Cancer Cohort Data Analysis 

Analysis of TCGA data was performed in R [54]. Statistical significance was defined as *p* < 0.05. Clinical information for patients with breast adenocarcinoma was retrieved from data published by Liu et al. [55]. The RNA sequencing data (FPKM) for *CDK7* and *ESR1* in primary tumor tissues from this cohort were retrieved from the GDC Data Portal [56]. A total of 981 primary breast tumors with clinical and mRNA data were included in the analysis, among them 715 ER+ tumors, 218 ER− tumors, and 48 with unknown ER status. The Spearman rank-order correlation test was applied to measure the strength of the association between mRNA expression levels of these two genes in all tumor samples and in ER+ samples. A univariate Cox proportional hazards model was fitted to evaluate the association between OS and covariates including *CDK7* expression and available clinical variables (age at diagnosis and disease stage). For the ER+ group, age and *CDK7* expression level were used to divide the cohort into low and high groups by the first and last sextiles, respectively. Age and *CDK7* status were statistically significant factors in the univariate Cox proportional hazards model and were included in the final multivariable analysis of OS. To visualize the survival difference, for each gene we used the log-rank test to find the point (cutoff) with the most significant (lowest *p-*value) split into high versus low mRNA level groups.

### 4.2. Tamoxifen-Treated Breast Cancer Cohort Data Analysis 

Analyses of the correlation of survival and mRNA expression were performed in R [54]. Statistical significance was defined as *p* < 0.05. We retrieved from the GEO repository microarray expression (normalized log_2_) data for *CDK7*, *ESR1*, *MYC*, and *SOX2* and OS information for 132 primary tumors from a study of tamoxifen-treated patients (GSE9893) [57]. The Spearman rank-order correlation test was applied to measure the strength of the association between mRNA expression levels of genes of interest. A univariate Cox proportional hazards model was fitted to evaluate the association between OS and covariates including mRNA expression and available clinical variables (age at diagnosis and local recurrence). To visualize the survival differences, for each gene we used a log-rank test to find the cutoff point with the lowest *p-*value to split the cohort into groups with high and low mRNA expression.

### 4.3. Cell Culture and Chemicals 

The human breast cancer cell lines MCF-7 (tamoxifen-sensitive) and LCC2 (tamoxifen-resistant counterpart of MCF-7) were provided by Robert Clarke (Georgetown University Medical Center, Washington, DC, USA) [58,59]. The LCC2 cells were developed after in vitro selection of the hormone-independent human breast cancer variant MCF-7/LCC1 that was stepwise incubated with triphenylethylene antiestrogen until it became resistant and produced the LCC2 variant. The LCC2 cells are superior to other resistant models because they are resistant to the inhibitory effects of 4-OHTAM but remain sensitive to the pure steroidal antiestrogen ICI 182,780 in vitro. They also have a tumorigenic potential *in vivo.* These advantages make LCC2 cells ideal model to investigate responsivity to antiestrogens and the development of cross-resistance both in vivo and *in vitro.* The MCF-7 cells were maintained in Dulbecco’s modified Eagle’s medium/F12 medium (Invitrogen, Grand Island, NY, USA) with 10% fetal bovine serum (FBS) (Thermo Scientific Hyclone, Rockford, IL) and 1% streptomycin (Sigma–Aldrich, St. Louis, MO, USA). The LCC2 cells were routinely maintained in minimum essential medium without phenol red and supplemented with 5% charcoal stripped FBS (Invitrogen) in a humidified 37 °C incubator containing 5% CO_2_. The LCC2 cells were tested for *Mycoplasma* using a MycoAlert detection kit (Lonza, Walkersville, MD, USA) before use, and the results were negative. Tamoxifen was purchased from Sigma–Aldrich and dissolved in dimethyl sulfoxide at a stock concentration of 10 mM. (*E*)-N-(3-(5-chloro-4-(1H-indol-3-yl)pyrimidin-2-ylamino)phenyl)-4-(4-(dimethylamino)but-2-enamido)benzamide (THZ1) was purchased from Millipore (Burlington, MA) and dissolved in dimethyl sulfoxide at stock concentration (10 μM).

### 4.4. siRNA Transfection

The MCF-7 and LCC2 cells were plated in six-well plates and seeded at density 2 × 10^5^/well. Twenty-four hours after seeding, cells reached 50%–60% confluence and were transfected with 25–200 nM siRNA targeting CDK7 for 72 h. Control plates included no treatment, and treatment with scramble non-targeting siRNA was transfected. Two different siRNAs (siRNA-CDK7-1 and siRNA-CDK7-2; Sigma–Aldrich) with the following using HiPerfect transfection reagent (Invitrogen) following protocols provided by the manufacturer. After 72 h, cells were collected for protein extraction or RNA isolation for RT-PCR. First, the two siRNA were studied to confirm its silencing effect on CDK7 using Western blot and RT-PCR with different concentrations.

### 4.5. RNA Isolation and RT-PCR 

The MCF-7 or LCC2 cells were seeded in six-well plates for 24 h and then treated with tamoxifen or/and THZ1. Forty-eight hours after treatment with tamoxifen and/or THZ1 and 72 h after transfection with CDK7 siRNA, the cells were typsinized and collected in tubes and then centrifuged for 5 min at 1200 rpm. The cellular total RNA was isolated from MCF-7 and LCC2 cells with Trizol reagent (Invitrogen), and cDNA was obtained from 1 μg of total RNA using a SuperScript II reverse transcription kit according to the manufacturer’s instructions (Invitrogen). The cDNA was then amplified by PCR with gene-specific primers for CDK7 and GAPDH on a CFX384 Touch Real-Time PCR Detection System (Bio-Rad Laboratories, Hercules, CA, USA) using SYBR Green PCR Master Mix (Promega) according to the manufacturer’s protocol. Fast amplification parameters were as follows: one cycle of 95 °C for 10 min, followed by 40 cycles of 95 °C for 15 s and 60 °C for 1 min. The primer sequences are shown in Table 1. Quantitative analysis of data was performed as described previously [60]. Values were normalized to GAPDH and are shown as relative expression levels.

### 4.6. Western Blotting 

The MCF-7 or LCC2 cells were seeded in sixx-well plates for 24 h and then treated with tamoxifen or/and THZ1. Forty-eight hours after treatment with tamoxifen and/or THZ1 and 72 h after transfection with CDK7 siRNA, the cells were typsinized and collected in tubes and then centrifuged for 5 min at 1200 rpm. Phosphate-buffered saline (PBS) was used to wash cells then lysed in 1 mL cell lysis buffer (pH 7.4; 10 mM Tris, 150 mM NaCl, 0.3% NP-40, 1% Triton X-100, 0.1 mmol/L EDTA, and complete proteinase inhibitor cocktail (Roche)) on ice for 10 min. Then, the lysates were put in the centrifuge, supernatants were taken, and protein concentration was measured using a Pierce BCA Protein Assay Kit (Thermo Fisher Scientific, Waltham, MA, USA). Each of the 40 µg protein samples was separated by electrophoresis through a 4% to 12% sodium dodecyl sulfate-polyacrylamide gel electrophoresis gel. The membrane (PVDF) was visualized by exposure to Kodak XAR film. All antibodies used are listed in Table 2. For the quantitative analysis, the mean intensity of each band (mean pixel) was compared with the β-Actin or GAPDH band using the ImagJ software developed by the National Institutes of Health and the Laboratory for Optical and Computational Instrumentation (LOCI, University of Wisconsin, Madison, WI, USA) [61].

### 4.7. SRB Assay 

The MCF-7 and LCC2 cells were seeded in each well of the 96 well plates at a density of 3000 cells/well and incubated for 24 h. Both cell lines were then treated with a range of concentrations of tamoxifen (10–80 μM) or THZ1 (1–100 nM) separately for 24 h. Next, both cell lines were treated with a range of concentrations of THZ1 (1–8 nM) combined with 1 or 17 µM of tamoxifen for different incubation times. Then, both cell lines were incubated with 1 or 17 µM of tamoxifen combined with 1 nM of THZ1 for 48 h. After that, cooled trichloroacetic acid (200 μL per well) was added and incubated for 60 min at 4°C. The plates were washed with distilled water and left to dry. The SRB solution (150 μL) at 0.4% w/v in 1% acetic acid was then added, and the plates were incubated for 30 min at room temperature. Following incubation, the plates were washed with 1% acetic acid and dried. Next, 100 μL of Tris base (10 mM) was put to the wells to solubilize the bound SRB, and absorbance was then read at 570 nm with an enzyme-linked immunosorbent assay microplate reader (Tecan Sunrise, Männedorf, Switzerland). The percentage of surviving cells was calculated using the following equation: (absorbance at wavelength 560 nM of test /absorbance at wavelength 560 nM of control) × 100%. The experiments were performed with at least three experimental replicates and three technical replicates. 

### 4.8. Quantification of Cell Death 

The MCF-7 and LCC2 cells were cultured in 96 well plates (1 × 10^4^ cells/well). The plates were incubated overnight at 37 °C, and, on the next day, 300 μL of culture media containing 1 µM of tamoxifen plus 1 nM of THZ1 or 17 µM of tamoxifen plus 1 nM of THZ1 were added to each well containing MCG-7 or LCC2 cells, respectively. The plates were then incubated at 37 °C in 5% CO_2_. After 48 h, 100 μL of medium from each well was carefully transferred to new plates. Next, 100 μL of LDH substrate was prepared according to the manufacturer’s instructions (Cytotoxicity Detection Kit, Roche Chemical Co., Basel, Switzerland) and added to each well. After 20 min of shaking at room temperature, the LDH activity was determined by measuring the change in absorbance at 490 nm with an enzyme-linked immunosorbent assay microplate reader (Tecan Sunrise). 

### 4.9. Annexin V/Propidium Iodide Staining for Apoptosis Assessment 

An Annexin-V-Fluos staining kit was used to detect apoptotic and necrotic cells according to the manufacturer’s instructions (Fisher Scientific, Waltham, MA, USA). This kit has double stains where, apoptotic cells and necrotic cells are stained with Annexin V (green fluorescence) and propidium iodide (PI, red fluorescence), respectively. Cells were grown to ∼70% confluence and treated with THZ1 and tamoxifen for 48 h or with CDK7 siRNA-2 for 72 h. After incubation, floating and adherent cells were collected and washed three times with cold PBS, then 5 μL of annexin V and PI was added. The cells were incubated for 20 min and then analyzed by FACScan (Becton Dickinson, Franklin Lakes, NJ, USA).

### 4.10. Cell Cycle Analysis

Cells were seeded at 70% confluence at 37 °C and 5% CO_2_. After 24 h, cells were treated with THZ1, tamoxifen for 48 h or CDK7 siRNA-2 for 72 h. After the treatment incubation, the floating cells were aspirated and discarded. The cells were detached using trypsin then washed two times with cold PBS, and centrifuged. The pellet was resuspended with PI for 20 min for flow cytometry analysis. Flow cytometry was performed with a FACScan flow cytometer (Becton Dickinson). A minimum of 10,000 cells/sample were collected, and cell cycle analysis was conducted with the resulting DNA histograms using ModiFitLT software (Verily Software House, Topsham, ME, USA).

### 4.11. Orthotopic Mouse Models of Breast Cancer 

We obtained five- to six-week-old female athymic nude mice (NCr nu/nu) from the Department of Experimental Radiation Oncology at The University of Texas MD Anderson Cancer Center (Houston, TX, USA). The mice were put in specific houses (5 per each) with standard acrylic glass cages in a room maintained at a recommended temperature and humidity with a 12 h light–dark cycle. They were supplied with a regular autoclaved chow diet with water ad libitum. For the experiments using MCF-7 and LCC2 cells, mice were primed with 17β-estradiol (Innovative Research of America, Sarasota, FL, USA) applied subcutaneously (0.32 mg estradiol/pellet) under the left shoulder to promote tumor growth. After 1 week, MCF-7 or LCC2 cells (6  ×  10^6^) were orthotopically injected into the right mammary fat pad of each mouse. When the tumors were 3 to 5 mm, mice were divided into four groups and administered no treatment, tamoxifen (1 mg/100 µL/mouse intraperitoneally every 2 days for 2 weeks) [62]), THZ1 (10 mg/kg^−1^ body weight intraperitoneally once per day for 1 week) [63], or the combination of tamoxifen and THZ1 at the dosages described above. After 5 weeks of treatment, the animals were humanely killed, and tumors were removed and processed for further experiments. Tumor weight and volume were recorded. Tumor tissue was fixed in formalin for paraffin embedding and frozen in optimal cutting temperature medium to prepare frozen slides or snap-frozen for lysate preparation.

### 4.12. Immunohistochemistry

Immunohistochemical analysis for Ki67 was performed on 4 μm formalin-fixed paraffin-embedded tumor sections. Slides were deparaffinized and dehydrated, then subjected to antigen retrieval using 1 × 530 Diva Decloaker (Biocare Medical, Pacheco, CA, USA) under a steamer. Hydrogen peroxide (3%) in methanol followed by washes with PBS was used to block endogenous peroxidases. Normal horse serum (5%) and normal goat serum (1%) in PBS was used to block normal binding. Samples were incubated with a primary antibody against Ki67 (1:200; Neomarkers, Portsmouth, NH, USA) overnight at 4 °C, and then a goat anti-rabbit horseradish peroxidase secondary antibody (Jackson ImmunoResearch Laboratories, West Grove, PA, USA) diluted in blocking solution was added. Immunohistochemical analyses for CD31 were performed on 8 μm sections of fresh frozen tumor specimens embedded in optimal cutting temperature medium. Slides were fixed with cold acetone and acetone–chloroform and rehydrated with PBS. Normal horse serum (5%) and normal goat serum (1%) in PBS was used to block nonspecific binding. Samples were incubated with the primary antibody, rat anti-mouse CD31 (1:200; BD Pharmingen, San Diego, CA, USA), overnight at 4 °C. Following washing the samples, a peroxidase-conjugated goat anti-rat secondary antibody (Jackson ImmunoResearch Laboratories) was added. with. The slides were incubated with 3,3’-diaminobenzidine (Sigma–Aldrich) at room temperature, counterstained with hematoxylin for 15 s, and mounted on a slide to be investigated on a bright-field microscope (magnification 20×). For TUNEL assays, 8 μm sections of fresh frozen tumor specimens embedded in optimal cutting temperature medium were used following the manufacturer’s instructions (Promega, Madison, WI, USA). The slides were examined under a fluorescent microscope (Nikon Eclipse TE2000-U, 20× magnification).

### 4.13. Statistical Analysis

All data are expressed as mean ± SD unless otherwise specified. Differences between treated samples and untreated controls were analyzed by one-way analysis of variance followed by the Tukey multiple comparison test. To test differences between untreated MCF-7 and LCC2 cells, unpaired *t*-tests were used. To test for interactions among individual treatments when given in combination, a factorial design test was used. Statistical analyses were performed using GraphPad Stat, version 7.03 (GraphPad, San Diego, CA, USA). The isobologram was used to determine the interaction between THZ1 and tamoxifen. Statistical significance was set at *p* < 0.05.

### 4.14. Ethical Approval

All animal studies were performed according to an experimental protocol approved by the MD Anderson Institutional Animal Care and Use Committee (approval code: 00001010-RN01 and 00001010-RN02, approved on 26 August 2019). 

## 5. Conclusions

This study was the first to show that targeting CDK7 (THZ1 or siRNA) induced tamoxifen cytotoxicity in sensitive and resistant breast cancer cell lines. Also, xenograft models showed a similar pattern to in vitro results, where THZ1 boosted the effect of tamoxifen on tumor weight and tumor volume. A combination of THZ1 and tamoxifen reduced the proliferation marker (Ki67) and angiogenesis markers (CD31) and increased apoptotic cells in xenograft models. In a clinical sample, CDK7 was associated with good response to tamoxifen in breast cancer patents. In a recent study, CDK7 enhanced tamoxifen’s effect by targeting the important transcription factor MYC, STAT3, and catenin pathways which paly roles in tamoxifen resistance.

## Figures and Tables

**Figure 1 ijms-21-02974-f001:**
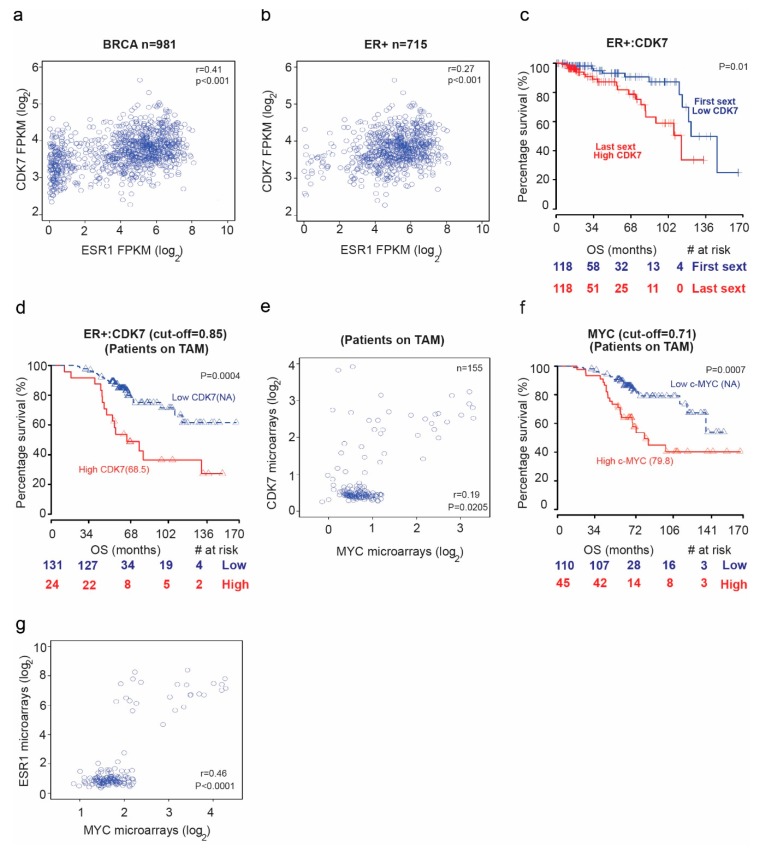
The relationship between cyclin dependent kinase (CDK7) expression and survival in breast cancer patients. Scatter plots of the Spearman rank-order correlation between (**a**) CDK7 and estrogen receptor alpha or 1 (ESR1) expression in 981 patients with breast cancer and (**b**) 715 patients with ER+ breast cancer. Data are from TCGA samples (RNASEqv2 data type). (**c**) Kaplan–Meier curves comparing overall survival (OS) in patients with ER+ breast cancer stratified by CDK7 expression level. (**d**) Kaplan–Meier curves comparing OS in patients with ER+ breast cancer receiving tamoxifen (TAM) by CDK7 expression level. (**e**) Scatter plot showing correlation between CDK7 and MYC expression in breast cancer patients receiving tamoxifen. (**f**) Kaplan–Meier curves comparing OS in breast cancer patients receiving tamoxifen by MYC expression level. (**g**) Scatter plot showing the correlation between MYC and ESR1 expression in breast cancer patients receiving tamoxifen.

**Figure 2 ijms-21-02974-f002:**
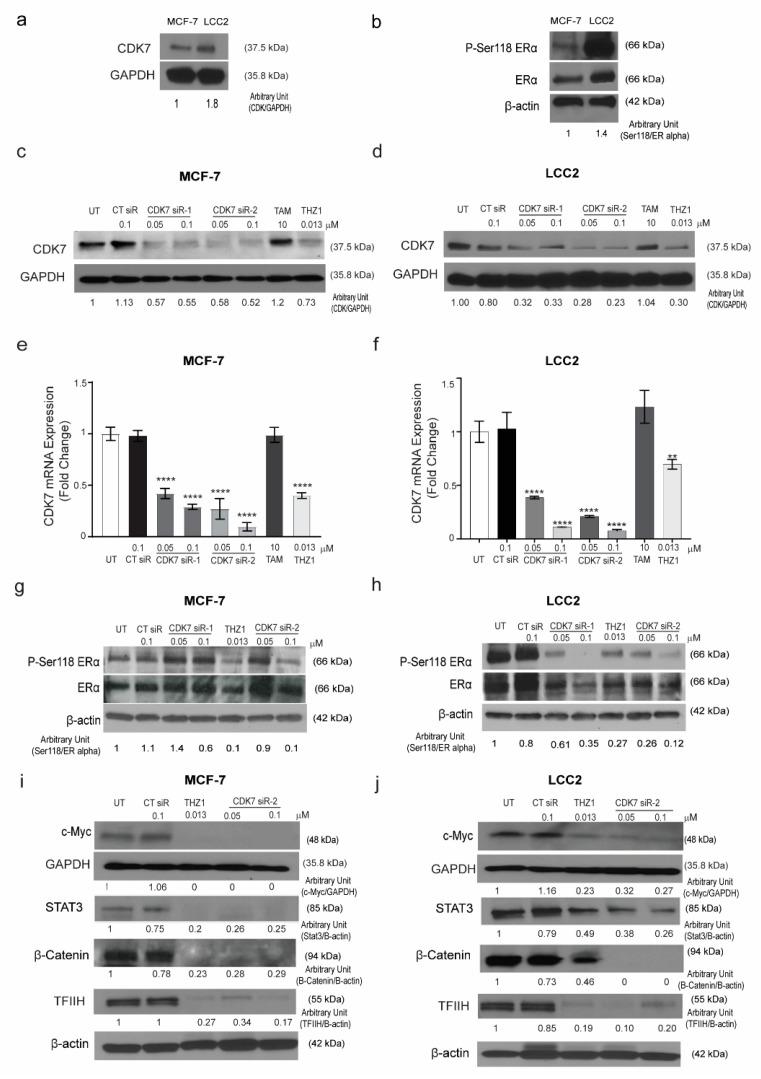
Expression of CDK7 in tamoxifen-sensitive and tamoxifen-resistant breast cancer cell lines. (**a**) Western blot showing CDK7 protein levels in tamoxifen-sensitive (MCF-7) and tamoxifen-resistant (LCC2) cells. (**b**) Western blots showing levels of ER-α and phosphorylated ER-α (at serine 118) (p-Ser118 ERα) in MCF-7 and LCC2 cells. (**c**,**d**) Western blots showing CDK7 protein levels in MCF-7 (**c**) and LCC2 (**d**) cells after transfection with 100 nM control siRNA (CT siR) or 50 or 100 nM siRNA-CDK7-1 (CDK7 siR-1) or siRNA-CDK7-2 (CDK7 siR-2) (72 h incubation) or treatment with 10 µM tamoxifen (TAM) or 13 nM THZ1 (48 h incubation). (**e**,**f**) Quantitative RT-PCR data showing CDK7 mRNA expression levels in MCF-7 (**e**) and LCC2 (**f**) cells after transfection or treatment as described above. The results are expressed as mean ± SD of five independent experiments performed in triplicate. Statistical significance was determined by one-way ANOVA using the Tukey multiple comparison test, ** *p* < 0.01, **** *p* < 0.001. (**g**,**h**) Western blots showing p-Ser118 ERα and ER-α levels in MCF-7 (**g**) and LCC2 (**h**) cells after transfection or treatment as described above. (**i,j**) Western blots showing levels of C-Myc, STAT3, β-catenin, and TFIIH in (**i**) MCF-7 and (**j**) LCC2 cells with no treatment after transfection with 100 nM CT siR, 50 or 100 nM siR-2, or 13 nM THZ1. For Western quantification imageJ software was used to measure the intensity and normalize each value to its corresponding β-actin or GAPDH.

**Figure 3 ijms-21-02974-f003:**
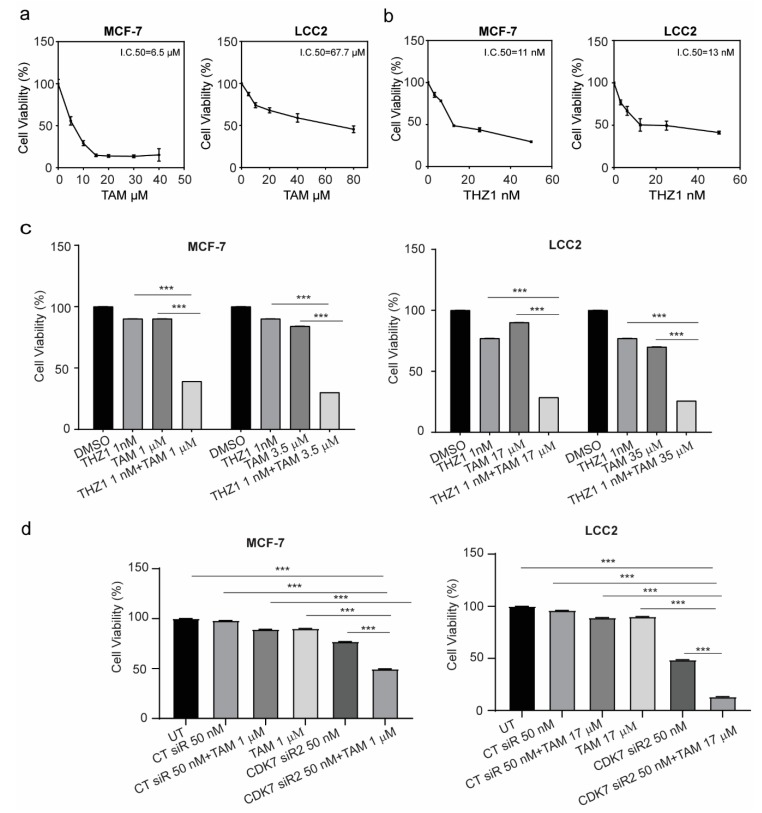
Viability of MCF-7 and LCC2 cells after tamoxifen, THZ1, or CDK7-targeting siRNA treatment individually and in combination. (**a**) Percentage of viable MCF-7 and LCC2 cells 48 h after treatment with the indicated concentrations of tamoxifen (TAM). (**b**) Percentage of viable MCF-7 and LCC2 cells after 48 h of incubation with THZ1 (2.5–50 nM). (**c**) (left) Viability of MCF-7 cells after treatment with the indicated concentrations of dimethyl sulfoxide (DMSO) (control), THZ1, tamoxifen, or THZ1 plus tamoxifen. (right) Viability of LCC2 cells after treatment with the indicated concentrations of dimethyl sulfoxide (DMSO), THZ1, tamoxifen, or THZ1 plus tamoxifen. (**d**) Cell viability after treatment with the indicated concentrations of control siRNA (CT siR) plus tamoxifen, siRNA-CDK7-2 (CDK7 siR), or siRNA-CDK7-2 plus tamoxifen in MCF-7 and LCC2 cells. The results are expressed as mean ± SD of five independent experiments performed in triplicate. Statistical significance was determined by one-way ANOVA using the Tukey multiple comparison test. *** *p* < 0.001. UT, untreated.

**Figure 4 ijms-21-02974-f004:**
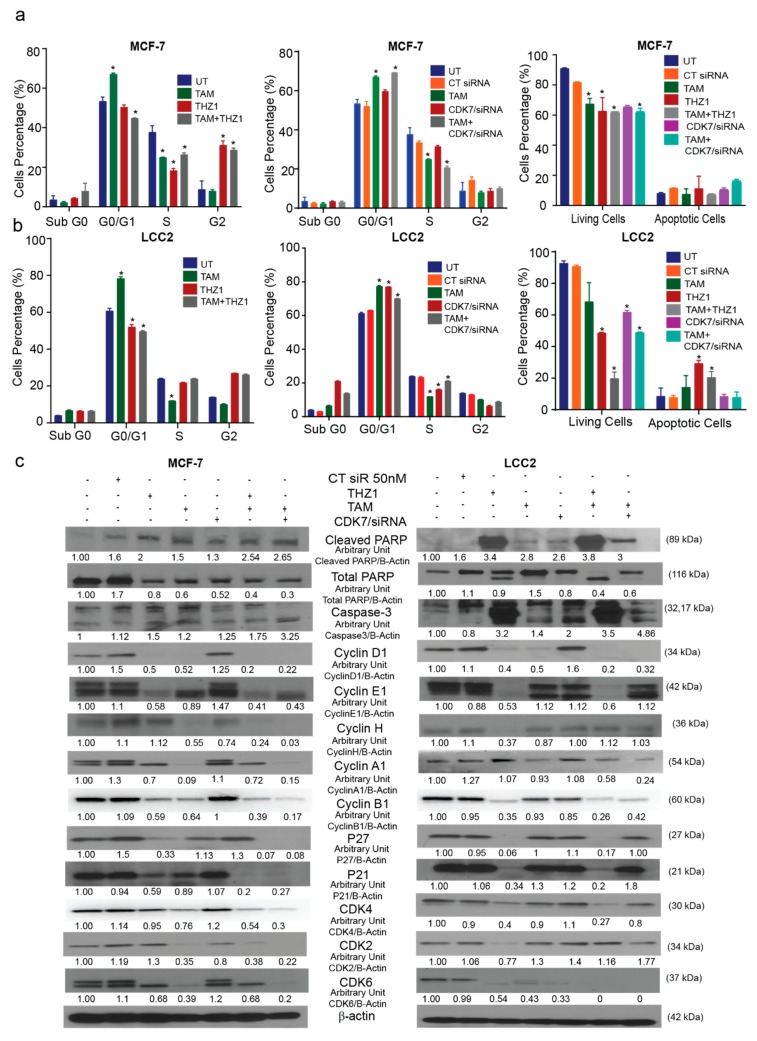
Effects of CDK7 inhibition combined with tamoxifen on the cell cycle and apoptosis. (**a**) Histograms showing the results of the flow cytometry analysis of the cell cycle and apoptosis assays of MCF-7 cells. (Left) Cell cycle analysis of MCF-7 cells treated with the indicated concentrations of tamoxifen, THZ1, and their combination. (Center) Cell cycle analysis of MCF-7 cells treated with the indicated concentrations of tamoxifen, CT siR, CDK7 siR, and the combination of tamoxifen plus CDK7 siR. (Right) Percentage of living and apoptotic MCF-7 cells treated with the indicated treatments. (**b**) Histograms showing results of flow cytometry analysis of the cell cycle and apoptosis assays of LCC2 cells. (Left) Cell cycle analysis of LCC2 cells treated with the indicated concentrations of tamoxifen, THZ1, and their combination. (Center) Cell cycle analysis of LCC2 cells treated with the indicated concentrations of tamoxifen CT siR, CDK7 siR, and the combination of tamoxifen plus CDK7 siR. (Right) Percentage of living and apoptotic LCC2 cells treated with the indicated treatments. (**c**) Western blots of cell cycle and apoptosis markers in cells exposed to tamoxifen, THZ1, and CDK7 siR, individually and in combination. For Western quantification imageJ software was used to measure the intensity and normalize each value to its corresponding β-actin. Results in the graphs are expressed as mean ± SD of two independent experiments performed in triplicate. Statistical significance was determined by one-way ANOVA using the Tukey multiple comparison test. * *p* < 0.05. UT, untreated.

**Figure 5 ijms-21-02974-f005:**
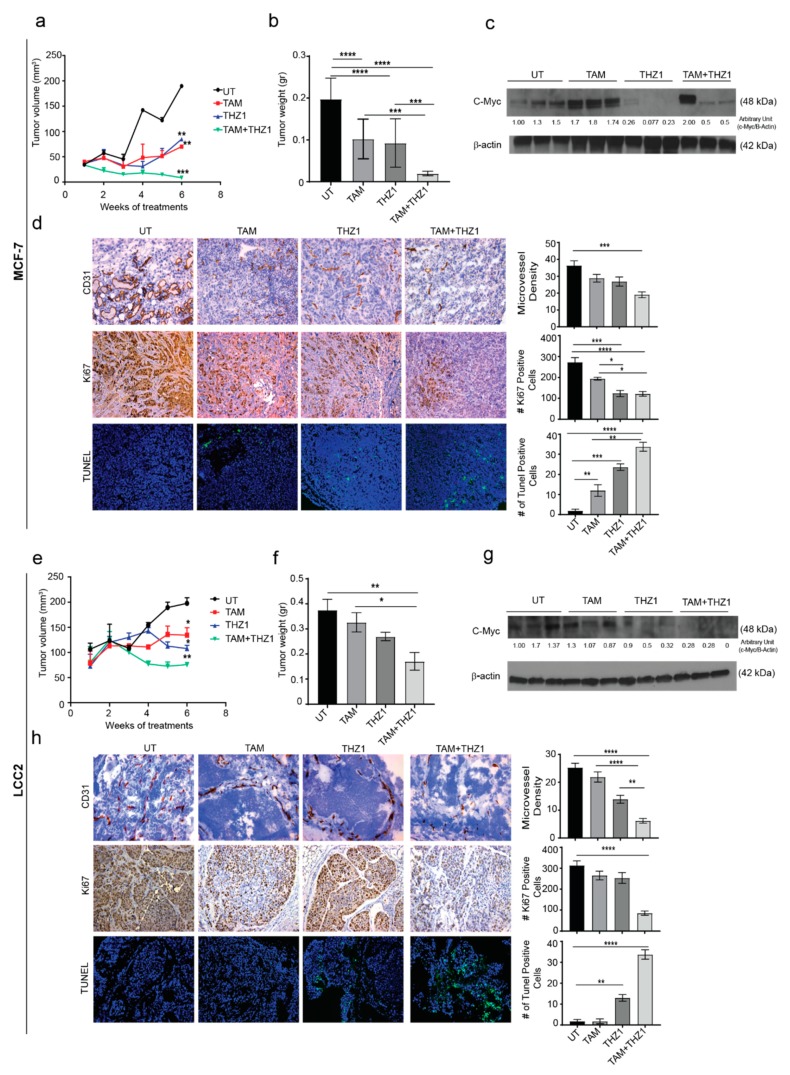
In vivo therapeutic efficacy of CDK7 inhibition in combination with tamoxifen. (**a**) Tumor volume in mice bearing MCF-7 tumors treated with tamoxifen, THZ1, or their combination. (**b**) Tumor weight in mice bearing MCF-7 tumors in four treatment groups (10 mice per group): untreated (UT), tamoxifen (TAM), THZ1, and tamoxifen plus THZ1. (**c**) Western blot of c-MYC protein levels in tumor tissue from mice bearing MCF-7 tumors treated with tamoxifen, THZ1, or their combination, for quantification we used imageJ software to measure the intensity and normalize each value to its corresponding β-actin. (**d**) (Left) Representative micrographs (20×) showing immunohistochemical staining of CD31 and Ki67 and terminal deoxynucleotidyl transferase dUTP nick end labeling (TUNEL) assays in MCF-7 tumors from mice treated with tamoxifen, THZ1, or their combination. (Right) Quantification of microvessel density, cell proliferation, and apoptosis in the above groups. (**e**) Tumor volume in mice bearing LCC2 tumors treated with tamoxifen, THZ1, or their combination. (**f**) Tumor weight in mice bearing LCC2 tumors in four treatment groups (10 mice per group): UT, tamoxifen, THZ1, and tamoxifen plus THZ1. (**g**) Western blot of c-MYC protein levels in tumor tissue from mice bearing LCC2 tumors treated with tamoxifen, THZ1, or their combination. (**h**) (Left) Representative micrographs (20 ×) showing immunohistochemical staining of CD31 and Ki67 and TUNEL assays in LCC2 tumors from mice treated with tamoxifen, THZ1, or their combination. (Right) Quantification of microvessel density, cell proliferation, and apoptosis in the above groups. Statistical significance was determined by one-way ANOVA using the Tukey multiple comparison test. **** *p* < 0.0001, *** *p* < 0.001, ** *p* < 0.01, * *p* < 0.05.

**Figure 6 ijms-21-02974-f006:**
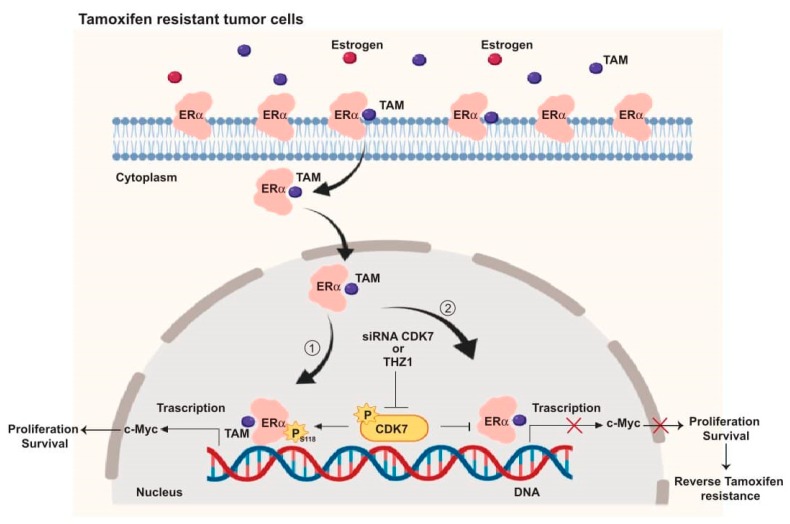
Schematic illustration of the CDK7 signaling pathway and how it interacts with ER and ER-regulating genes including MYC. ① CDK7 activates ER phosphorylation at serine 118, which augments MYC transcription. ② Targeting CDK7 by siRNA or THZ1 blocks ER phosphorylation and MYC expression which could restore tamoxifen (TAM) toxicity in resistant breast cancers.

**Table 1 ijms-21-02974-t001:** Genes sequences.

Gene Symbol	Forward Sequence	Reverse Sequence
CDK7	5′-AGGATGTATGGTGTAGGTGTGGA-3′	5′-AAGATGTGATGCAAAGGTATTCC-3′
GAPDH	5′-TGAAGGTCGGAGTCAACGGATTT-3′	5′-GCCATGGAATTTGCCATGGGTGG-3′

**Table 2 ijms-21-02974-t002:** Antibodies information details.

Primary Antibody	Purchasing Company	Dilution Factor	Secondary Company	Purchasing Company	Dilution Factor
Anti-estrogen receptor alpha	Santa-Cruz biotechnology, Dallas, TX, USA	1:100	Horseradish peroxidase (HRP) linked secondary antibody Anti-mouse	Sigma–Aldrich, St. Louis, MO, USA	1:2000
Anti-Ser118	Cell signal Technology, Danvers, MA, USA	1:1000	HRP linked secondary antibody Anti-rabbit	Sigma–Aldrich, St. Louis, MO, USA	1:2000
Anti-Stat3	Cell signal Technology, Danvers, MA, USA	1:1000	HRP linked secondary antibody Anti-rabbit	Sigma–Aldrich, St. Louis, MO, USA	1:2000
Anti-β-catenin	Santa-Cruz biotechnology, Dallas, TX, USA	1:100	HRP linked secondary antibody Anti-mouse	Sigma–Aldrich, St. Louis, MO, USA	1:2000
Anti-CDK 2	Cell signal Technology, Danvers, MA, USA	1:1000	HRP linked secondary antibody Anti-mouse	Sigma–Aldrich, St. Louis, MO, USA	1:2000
Anti-CDK 4	Cell signal Technology, Danvers, MA, USA	1:1000	HRP linked secondary antibody Anti-rabbit	Sigma–Aldrich, St. Louis, MO, USA	1:2000
Anti-CDK 6	Cell signal Technology, Danvers, MA, USA	1:1000	HRP linked secondary antibody Anti-rabbit	Sigma–Aldrich, St. Louis, MO, USA	1:2000
Anti-CDK 7	Cell signal Technology, Danvers, MA, USA	1:1000	HRP linked secondary antibody Anti-mouse	Sigma–Aldrich, St. Louis, MO, USA	1:2000
Anti-cyclin A1	Santa-Cruz biotechnology, Dallas, TX, USA	1:100	HRP linked secondary antibody Anti-mouse	Sigma–Aldrich, St. Louis, MO, USA	1:2000
Anti-cyclin B1	Cell signal Technology, Danvers, MA, USA	1:1000	HRP linked secondary antibody Anti-rabbit	Sigma–Aldrich, St. Louis, MO, USA	1:2000
Anti-cyclin H	Cell signal Technology, Danvers, MA, USA	1:1000	HRP linked secondary antibody Anti-rabbit	Sigma–Aldrich, St. Louis, MO, USA	1:2000
Anti-cyclin D1	Cell signal Technology, Danvers, MA, USA	1:1000	HRP linked secondary antibody Anti-rabbit	Sigma–Aldrich, St. Louis, MO, USA	1:2000
Anti-cyclin E1	Cell signal Technology, Danvers, MA, USA	1:1000	HRP linked secondary antibody Anti-rabbit	Sigma–Aldrich, St. Louis, MO, USA	1:2000
Anti-p21	Santa-Cruz biotechnology, Dallas, TX, USA	1:100	HRP linked secondary antibody Anti-mouse	Sigma–Aldrich, St. Louis, MO, USA	1:2000
Anti-p27	Santa-Cruz biotechnology, Dallas, TX, USA	1:100	HRP linked secondary antibody Anti-mouse		1:2000
Caspase-3	Sigma–Aldrich, St. Louis, MO, USA	1:1000	HRP linked secondary antibody Anti-rabbit	Sigma–Aldrich, St. Louis, MO, USA	1:2000
Cleaved PARP	Cell signal Technology, Danvers, MA, USA	1:1000	HRP linked secondary antibody Anti-rabbit	Sigma–Aldrich, St. Louis, MO, USA	1:2000
PARP	Cell signal Technology, Danvers, MA, USA	1:1000	HRP linked secondary antibody Anti-rabbit	Sigma–Aldrich, St. Louis, MO, USA	1:2000
C-Myc	Cell signal Technology, Danvers, MA, USA	1:1000	HRP linked secondary antibody Anti-rabbit	Sigma–Aldrich, St. Louis, MO, USA	1:2000

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
