# Peer review of "Blockade of CDK7 Reverses Endocrine Therapy Resistance in Breast Cancer"

_ijms, 2020, doi:10.3390/ijms21082974_

Round 1
Reviewer 1 Report
In the present manuscript, Attia et al. provide evidence of a role for CDK7 as a potential therapeutic target for breast cancers, especially in tamoxifen-resistant setting. The authors show that high CDK7 expression is associated with shorter overall survival and tamoxifen response in ER-positive breast cancer patients. Moreover, inhibition of CDK7 reduced proliferation and increased apoptosis in ‘in vitro’ and ‘in vivo’ tamoxifen-sensitive and -resistant breast cancer models. The manuscript is well written and data support the conclusion; however, additional points should be covered:
- Immunoblotting: i) images are not of quality publication (see Fig 2G and H, some panels of Fig 4C, please improve their quality); ii) all loading controls are overexposed (please substitute them); iii) densitometric analysis should be provided for all of the bands of interest;
- It seems that THZ1, in addition to affect CDK7 activity, is able to affect CDK7 expression (Fig. 2C and D), please provide other references showing modulation in CDK7 protein expression;
- 3 and 4: it should interesting to evaluate whether Tam at lower concentration (1μM) in combination with THZ1 or CDK7 siR2 could be effective in affecting cell viability and apoptosis of LCC2 cells; most probably, there is a mistake in the horizontal axis of Fig 3C right panel regarding Tam concentration;
- A speculation on the different effects shown for MCF-7 and LCC2 on cell cycle should be done in the ‘Discussion’ section;
- Fig 5: why did Tamoxifen increase c-MYC expression in parental cells? Please discuss it. In addition, pSer118 ERα should be shown in tumor xenograft lysates under the different experimental conditions;
- Please briefly specify in ‘Material and Method’ section, the condition of selection of LCC2 cells.
Reviewer 2 Report
Authors demonstrated that CDK7 expression is associated with shorter overall survival and tamoxifen response in ER+ breast cancer and that CDK7 promotes multiple pro-oncogenic pathways important in the regulation of tumor progression and tamoxifen resistance. Their finding that CDK7 regulates MYC is not novel (see 10.3390/cells9030638) but important. In vivo inhibition of CDK7 increased apoptosis, reduced angiogenesis, and inhibited tumor growth in tamoxifen-sensitive and -resistant orthotopic mouse models of breast cancer. They suggested CDK7 as a possible therapeutic target for breast cancers that are resistant to tamoxifen therapy.
The data seems to be mostly clear and the manuscript is logically organized, there are several points to be improved.
Page1
19-22: Methods chapter of the abstract does not describe the actual experimental methods used in the manuscript, authors using incomplete sentences.
34: Please spell out all abbreviations the first time they are used in the text (despite the fact that they were mentioned in the abstract). Grammar: females, or female patients - should be plural
Page 2
8-9: CDK7 is involved in both the cell cycle and transcription regulation, as a CDK-activating kinase (CAK) and as a component of the general transcription factor TFIIH, respectively. Paper cited here lists 21 CDKs, and clearly indicates that CDK7 isn't directly involved in cell cycle progression. Additional papers for more comprehensive review (doi: 10.1158/0008-5472, doi:10.3390/cells9030638, doi: 10.1080/21541264.2018.1553483)
36-37: P values are significant, but R values seem low, please justify.
41: ER+ status or ESR1 expression? The legend for Fig. 1g states that it was ESR1 expression.
Page 4
2: GAPDH expression is very high and difference looks minimal/hard to judge because of it. Quantification would help convince me that this is true.
6: Same. Quantification would help.
12-13: make sure to make clear how long was THZ1/tamoxifen treatment for WB and RT-PCR samples. In figure 2 legend, but NOT in methods section.
Fig 2 (e-f) Fold change compared to what? Mean of untreated in LCC2 is more than 1, were untreated set in parental MCF7 to baseline? Needs to be clarified.
Page 6
11-15: Should be noted that 17microM dose is based on the MCF7 cell data (and spelled out when “half IC50” mentioned in line 10) and 35microM dose is based on LCC2 data for clarity. Also, why was 1microM/3.5microM TAM + 1nM THZ1 not tried in LCC2 cells for direct comparison with MCF7 response?
Page 8
4-9: WB - No noticeable differences in MCF7 cells. In LCC2 cells, THZ1 alone has a huge effect, completely overwhelming any contribution of TAM when treated in combination. Different concentrations should be used to get clear data. siRNA effect is more convincing in LCC2, but still modest.
Why different from earlier experiments THZ1 concentration was used? THZ1 is too high here.
Also, Tamoxifen alone and THZ1 alone decrease Cyclin D expression to nearly nothing, so how can the authors make any statement about increased effect in combination?
Page 10
8-12: This paragraph/sentence seems a bit jumbled up. Suggest splitting in 2-3 sentences.
17: Except in the one sample where the combo strongly increased MYC expression.
47: More accurate to say "overcomes" rather that “reverses”
Page 12:
The discussion is fine content-wise, but is wordy and a bit disorganized. Additional papers were published and might help to reinforce the discussion: doi:https://doi.org/10.1016/j.cell.2014.10.024, doi: 10.1158/0008-5472, doi:10.3390/cells9030638.
Round 2
Reviewer 1 Report
Authors have attempted to reply to the reviewer’s criticiscms, although it was not easy to determine since there was no correspondence about lines/pages indicated in the cover letter and the manuscript that it can be downloaded. The answers can be considered satisfactory. However, still minor concerns about densitometric analysis should be addressed; in particolar: i) in the figure legends, it lacks any description of what numbers under the blots may represent; ii) numbers should indicate the mean of the band optical density of the protein of interest vs loading/transfer control expressed as a fold over control of the specific experiment (which will be assumed to be 1) or a percentage over control of the specific experiment (which will be assumed to be 100%); iii) how were the bands of interest quantified? There is no description of densitometry scanning program in the ‘Material and Methods’ section.
Reviewer 2 Report
Authors answered this reviewer's comments and made requested changes in the manuscript. Updated Figure 4 is improved, although longer exposure would make effect of THZ1/TAM combination on Cyclin D confirmed better, especially in MCF7 cells. Figures 2 and 4 are somewhat overcrowded; explanation how normalization and quantification were performed may well be included in the figure legend.
